# Generating and Exploiting Large-scale Pseudo Training Data for Zero Pronoun Resolution

## Abstract

Most existing approaches for zero pronoun resolution are heavily relying on annotated data, which is often released by shared task organizers. Therefore, the lack of annotated data becomes a major obstacle in the progress of zero pronoun resolution task. Also, it is expensive to spend manpower on labeling the data for better performance. To alleviate the problem above, in this paper, we propose a simple but novel approach to automatically generate large-scale pseudo training data for zero pronoun resolution. Furthermore, we successfully transfer the cloze-style reading comprehension neural network model into zero pronoun resolution task and propose a two-step training mechanism to overcome the gap between the pseudo training data and the real one. Experimental results show that the proposed approach significantly outperforms the state-of-the-art systems with an absolute improvements of 3.1% F-score on OntoNotes 5.0 data.

## 1 Introduction

Previous works on zero pronoun (ZP) resolution mainly focused on the supervised learning approaches (Han, 2006; Zhao and Ng, 2007; Iida et al., 2007; Kong and Zhou, 2010; Iida and Poesio, 2011; Chen and Ng, 2013). However, a major obstacle for training the supervised learning models for ZP resolution is the lack of annotated data. An important step is to organize the shared task on anaphora and coreference resolution, such as the ACE evaluations, SemEval-2010 shared task on Coreference Resolution in Multiple Languages (Marta Recasens, 2010) and CoNLL-2012 shared task on Modeling Multilingual Unrestricted Coreference in OntoNotes (Sameer Pradhan, 2012). Following these shared tasks, the annotated evaluation data can be released for the following researches. Despite the success and contributions of these shared tasks, it still faces the challenge of spending manpower on labeling the extended data for better training performance and domain adaptation.

To address the problem above, in this paper, we propose a simple but novel approach to automatically generate large-scale pseudo training data for zero pronoun resolution. Inspired by data generation on cloze-style reading comprehension, we can treat the zero pronoun resolution task as a special case of reading comprehension problem. So we can adopt similar data generation methods of reading comprehension to the zero pronoun resolution task. For the noun or pronoun in the document, which has the frequency equal to or greater than 2, we randomly choose one position where the noun or pronoun is located on, and replace it with a specific symbol $\langle blank \rangle$. Let query $\mathcal{Q}$ and answer $\mathcal{A}$ denote the sentence that contains a $\langle blank \rangle$, and the noun or pronoun which is replaced by the $\langle blank \rangle$, respectively. Thus, a pseudo training sample can be represented as a triple:

$$\langle \mathcal{D}, \mathcal{Q}, \mathcal{A} \rangle \qquad (1)$$

For the zero pronoun resolution task, a $\langle blank \rangle$ represents a zero pronoun (ZP) in query $\mathcal{Q}$, and $\mathcal{A}$ indicates the corresponding antecedent of the ZP. In this way, tremendous pseudo training samples can be generated from the various documents, such as news corpus.

Towards the shortcomings of the previous approaches that are based on feature engineering, we propose a neural network architecture, which is an attention-based neural network model, for zero pronoun resolution. Also we propose a two-step

training method, which benefit from both large-scale pseudo training data and task-specific data, showing promising performance.

To sum up, the contributions of this paper are listed as follows.

- To our knowledge, this is the first time that utilizing reading comprehension neural network model into zero pronoun resolution task.

- We propose a two-step training approach, namely pre-training-then-adaptation, which benefits from both the large-scale automatically generated pseudo training data and task-specific data.

- Towards the shortcomings of the feature engineering approaches, we first propose an attention-based neural network model for zero pronoun resolution.

## 2 The Proposed Approach

In this section, we will describe our approach in detail. First, we will describe our method of generating large-scale pseudo training data for zero pronoun resolution. Then we will introduce two-step training approach to alleviate the gaps between pseudo and real training data. Finally, the attention-based neural network model as well as associated unknown words processing techniques will be described.

### 2.1 Generating Pseudo Training Data

In order to get large quantities of training data for neural network model, we propose an approach, which is inspired by (Hermann et al., 2015), to automatically generate large-scale pseudo training data for zero pronoun resolution. However, our approach is much more simple and general than that of (Hermann et al., 2015). We will introduce the details of generating the pseudo training data for zero pronoun resolution as follows.

First, we collect a large number of documents that are relevant (or homogenous in some sense) to the released OntoNote 5.0 data for zero pronoun resolution task in terms of its domain. In our experiments, we used large-scale news data for training.

Given a certain document $\mathcal{D}$, which is composed by a set of sentences $\mathcal{D} = \{s_1, s_2, ..., s_n\}$, we randomly choose an answer word $\mathcal{A}$ in the document. Note that, we restrict $\mathcal{A}$ to be either a noun or pronoun, where the part-of-speech is identified using LTP Toolkit (Che et al., 2010), as well as the answer word should appear at least twice in the document. Second, after the answer word $\mathcal{A}$ is chosen, the sentence that contains $\mathcal{A}$ is defined as a query $\mathcal{Q}$, in which the answer word $\mathcal{A}$ is replaced by a specific symbol $\langle blank \rangle$. In this way, given the query $\mathcal{Q}$ and document $\mathcal{D}$, the target of the prediction is to recover the answer $\mathcal{A}$. That is quite similar to the zero pronoun resolution task. Therefore, the automatically generated training samples is called **pseudo** training data. Figure 1 shows an example of a pseudo training sample.

In this way, we can generate tremendous triples of $\langle \mathcal{D}, \mathcal{Q}, \mathcal{A} \rangle$ for training neural network, without making any assumptions on the nature of the original corpus.

### 2.2 Two-step Training

It should be noted that, though we have generated large-scale pseudo training data for neural network training, there is still a gap between pseudo training data and the real zero pronoun resolution task in terms of the query style. So we should do some adaptations to our model to deal with the zero pronoun resolution problems ideally.

In this paper, we used an effective approach to deal with the mismatch between pseudo training data and zero pronoun resolution task-specific data. Generally speaking, in the first stage, we use a large amount of the pseudo training data to train a fundamental model, and choose the best model according to the validation accuracy. Then we continue to train from the previous best model using the zero pronoun resolution task-specific training data, which is exactly the same domain and query type as the standard zero pronoun resolution task data.

The using of the combination of proposed pseudo training data and task-specific data, i.e. zero pronoun resolution task data, is far more effective than using either of them alone. Though there is a gap between these two data, they share many similar characteristics to each other as illustrated in the previous part, so it is promising to utilize these two types of data together, which will compensate to each other.

The two-step training procedure can be concluded as,

```
Document:
1 |||    welcome both of you to the studio to participate in our program ,
        欢迎 两位 呢 来 演播室 参与 我们 的 节目 ,
2 |||    it happened that i was going to have lunch with a friend at noon .
        正好 因为 我 也 和 朋友 这个 , 这个 中午 一起 吃饭 。
3 |||    after that , i received an sms from 1860 .
        然后 我 就 收到 1860 的 短信 。
4 |||    uh-huh , it was by sms .
        嗯 , 是 通过 短信 的 方式 ,
5 |||    uh-huh , that means , er , you knew about the accident through the source of radio station .
        嗯 , 就是说 呃 你 是 通过 台 里面 的 一个 信息 的 渠道 知道 这儿 出 了 这样 的 事故 。
6 |||    although we live in the west instead of the east part , and it did not affect us that much ,
        虽然 我们 生活 在 西部 不 是 在 东部 , 对 我们 影响 不 是 很 大 ,
7 |||    but i think it is very useful to inform people using sms .
        但是 呢 , 我 觉得 有 这样 一个 短信 告诉 大家 呢 是 非常 有用 的 啊 。
Query:
8 |||    some car owners said that <blank> was very good。
        有 车主 表示 , 说 这 <blank> 非常 的 好。
Answer:
sms
短信
```

Figure 1: Example of pseudo training sample for zero pronoun resolution system. (The original data is in Chinese, we translate this sample into English for clarity)

- Pre-training stage: by using large-scale training data to train the neural network model, we can learn richer word embeddings, as well as relatively reasonable weights in neural networks than just training with a small amount of zero pronoun resolution task training data;

- Adaptation stage: after getting the best model that is produced in the previous step, we continue to train the model with task-specific data, which can force the previous model to adapt to the new data, without losing much knowledge that has learned in the previous stage (such as word embeddings).

As we will see in the experiment section that the proposed two-step training approach is effective and brings significant improvements.

### 2.3 Attention-based Neural Network Model

Our model is primarily an attention-based neural network model, which is similar to *Attentive Reader* proposed by (Hermann et al., 2015). Formally, when given a set of training triple $\langle \mathcal{D}, \mathcal{Q}, \mathcal{A} \rangle$, we will construct our network in the following way.

Firstly, we project one-hot representation of document $\mathcal{D}$ and query $\mathcal{Q}$ into a continuous space with the shared embedding matrix $W_e$. Then we input these embeddings into different bi-directional RNN to get their contextual representations respectively. In our model, we used the bidirectional Gated Recurrent Unit (GRU) as RNN implementation (Cho et al., 2014).

$$e(x) = W_e \cdot x, \ where \ x \in \mathcal{D}, \mathcal{Q} \quad (2)$$

$$\overrightarrow{h_s} = \overrightarrow{GRU}(e(x)); \overleftarrow{h_s} = \overleftarrow{GRU}(e(x)) \quad (3)$$

$$h_s = [\overrightarrow{h_s}; \overleftarrow{h_s}] \quad (4)$$

For the query representation, instead of concatenating the final forward and backward states as its representations, we directly get an averaged representations on all bi-directional RNN slices, which can be illustrated as

$$h_{query} = \frac{1}{n} \sum_{t=1}^{n} h_{query}(t) \quad (5)$$

For the document, we place a soft attention over all words in document (Bahdanau et al., 2014), which indicate the degree to which part of document is attended when filling the blank in the query sentence. Then we calculate a weighted sum of all document tokens to get the attended representation of document.

$$m(t) = \tanh(W \cdot h_{doc}(t) + U \cdot h_{query}) \quad (6)$$

$$\alpha(t) = \frac{\exp(W_s \cdot m(t))}{\sum_{j=1}^{n} \exp(W_s \cdot m(j))} \quad (7)$$

$$h_{doc\_att} = h_{doc} \cdot \alpha \quad (8)$$

where variable $\alpha(t)$ is the normalized attention weight at $t$th word in document, $h_{doc}$ is a matrix that concatenate all $h_{doc}(t)$ in sequence.

$$h_{doc} = concat[h_{doc}(1), h_{doc}(2), ..., h_{doc}(t)] \quad (9)$$

Then we use attended document representation and query representation to estimate the final answer, which can be illustrated as follows, where $V$

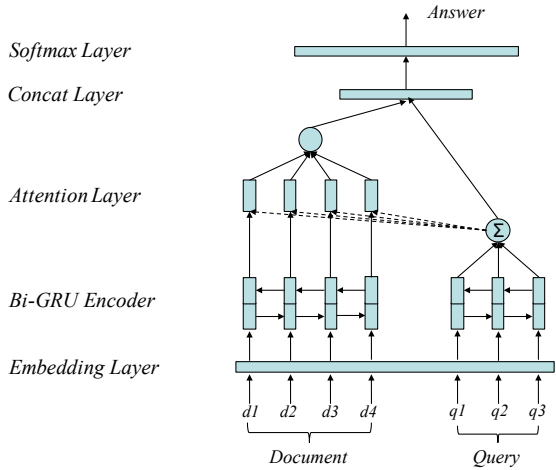

Figure 2: Architecture of attention-based neural network model for zero pronoun resolution task.

is the vocabulary,

$$r = concat[h_{doc\_att}, h_{query}] \qquad (10)$$

$$P(\mathcal{A}|\mathcal{D}, \mathcal{Q}) \propto softmax(W_r \cdot r) \ , s.t. \ \mathcal{A} \in V \qquad (11)$$

Figure 2 shows the proposed neural network architecture.

Note that, for zero pronoun resolution task, antecedents of zero pronouns are always noun phrases (NPs), while our model generates only one word (a noun or a pronoun) as the result. To better adapt our model to zero pronoun resolution task, we further process the output result in the following procedure. First, for a given zero pronoun, we extract a set of NPs as its candidates utilizing the same strategy as (Chen and Ng, 2015). Then, we use our model to generate an answer (one word) for the zero pronoun. After that, we go through all the candidates from the nearest to the far-most. For an NP candidate, if the produced answer is its head word, we then regard this NP as the antecedent of the given zero pronoun. By doing so, for a given zero pronoun, we generate an NP as the prediction of its antecedent.

## 2.4 Unknown Words Processing

Because of the restriction on both memory occupation and training time, it is usually suggested to use a shortlist of vocabulary in neural network training. However, we often replace the out-of-vocabularies to a unique special token, such as $\langle unk \rangle$. But this may place an obstacle in real world test. When the model predicts the answer as $\langle unk \rangle$, we do not know what is the exact word it represents in the document, as there may have many $\langle unk \rangle$s in the document.

In this paper, we propose to use a simple but effective way to handle unknown words issue. The idea is straightforward, which can be illustrated as follows.

- Identify all unknown words inside of each $\langle \mathcal{D}, \mathcal{Q}, \mathcal{A} \rangle$;

- Instead of replacing all these unknown words into one unique token $\langle unk \rangle$, we make a hash table to project these unique unknown words to numbered tokens, such as $\langle unk1 \rangle, \langle unk2 \rangle, ..., \langle unkN \rangle$ in terms of its occurrence order in the document. Note that, the same words are projected to the same unknown word tokens, and all these projections are only valid inside of current sample. For example, $\langle unk1 \rangle$ indicate the first unknown word, say "apple", in the current sample, but in another sample the $\langle unk1 \rangle$ may indicate the unknown word "orange". That is, the unknown word labels are indicating position features rather than the exact word;

- Insert these unknown marks in the vocabulary. These marks may only take up dozens of slots, which is negligible to the size of shortlists (usually 30K $\sim$ 100K).

(a) *The weather today is not as pleasant as the weather of yesterday.*
(b) *The **<unk>** today is not as **<unk>** as the **<unk>** of yesterday.*
(c) *The **<unk1>** today is not as **<unk2>** as the **<unk1>** of yesterday.*

Figure 3: Example of unknown words processing. a) original sentence; b) original unknown words processing method; c) our method

We take one sentence "The weather of today is not as pleasant as the weather of yesterday." as an example to show our unknown word processing method, which is shown in Figure 3.

If we do not discriminate the unknown words and assign different unknown words with the same token $\langle unk \rangle$, it would be impossible for us to know what is the exact word that $\langle unk \rangle$ represents for in the real test. However, when using our proposed unknown word processing method, if the model predicts a $\langle unkX \rangle$ as the answer,

we can simply scan through the original document and identify its position according to its unknown word number $X$ and replace the $\langle unkX \rangle$ with the real word. For example, in Figure 3, if we adopt original unknown words processing method, we could not know whether the $\langle unk \rangle$ is the word "weather" or "pleasant". However, when using our approach, if the model predicts an answer as $\langle unk1 \rangle$, from the original text, we can know that $\langle unk1 \rangle$ represents the word "weather".

## 3 Experiments

### 3.1 Data

In our experiments, we choose a selection of public news data to generate large-scale pseudo training data for pre-training our neural network model (pre-training step)[1]. In the adaptation step, we used the official dataset OntoNotes Release 5.0[2] which is provided by CoNLL-2012 shared task, to carry out our experiments. The CoNLL-2012 shared task dataset consists of three parts: a training set, a development set and a test set. The datasets are made up of 6 different domains, namely Broadcast News (BN), Newswires (NW), Broadcast Conversations (BC), Telephone Conversations (TC), Web Blogs (WB), and Magazines (MZ). We closely follow the experimental settings as (Kong and Zhou, 2010; Chen and Ng, 2014, 2015, 2016), where we treat the training set for training and the development set for testing, because only the training and development set are annotated with ZPs. The statistics of training and testing data is shown in Table 1 and 2 respectively.

|  | Sentences # | Query # |
|---|---|---|
| General Train | 18.47M | 1.81M |
| Domain Train | 122.8K | 9.4K |
| Validation | 11,191 | 2,667 |

Table 1: Statistics of training data, including pseudo training data and OntoNotes 5.0 training data.

### 3.2 Neural Network Setups

Training details of our neural network models are listed as follows.

---

[1] The news data is available at http://www.sogou.com/labs/dl/cs.html

[2] http://catalog.ldc.upenn.edu/LDC2013T19

|  | Docs | Sentences | Words | AZPs |
|---|---|---|---|---|
| Test | 172 | 6,083 | 110K | 1,713 |

Table 2: Statistics of test set (OntoNotes 5.0 development data).

- **Embedding:** We use randomly initialized embedding matrix with uniformed distribution in the interval [-0.1,0.1], and set units number as 256. No pre-trained word embeddings are used.

- **Hidden Layer:** We use GRU with 256 units, and initialize the internal matrix by random orthogonal matrices (Saxe et al., 2013). As GRU still suffers from the gradient exploding problem, we set gradient clipping threshold to 10.

- **Vocabulary:** As the whole vocabulary is very large (over 800K), we set a shortlist of 100K according to the word frequency and unknown words are mapped to 20 $\langle unkX \rangle$ using the proposed method.

- **Optimization:** We used ADAM update rule (Kingma and Ba, 2014) with an initial learning rate of 0.001, and used negative log-likelihood as the training objective. The batch size is set to 32.

All models are trained on Tesla K40 GPU. Our model is implemented with Theano (Theano Development Team, 2016) and Keras (Chollet, 2015).

### 3.3 Experimental results

Same to the previous researches that are related to zero pronoun resolution, we evaluate our system performance in terms of F-score (F). We focus on AZP resolution process, where we assume that gold AZPs and gold parse trees are given[3]. The same experimental setting is utilized in (Chen and Ng, 2014, 2015, 2016). The overall results are shown in Table 3, where the performances of each domain are listed in detail and overall performance is also shown in the last column.

- **Overall Performance**

We employ four Chinese ZP resolution baseline systems on OntoNotes 5.0 dataset. As we can

---

[3] All gold information are provided by the CoNLL-2012 shared task dataset

|  | NW (84) | MZ (162) | WB (284) | BN (390) | BC (510) | TC (283) | **Overall** |
|---|---|---|---|---|---|---|---|
| Kong and Zhou (2010) | 34.5 | 32.7 | 45.4 | 51.0 | 43.5 | 48.4 | 44.9 |
| Chen and Ng (2014) | 38.1 | 31.0 | 50.4 | 45.9 | 53.8 | **54.9** | 48.7 |
| Chen and Ng (2015) | 46.4 | 39.0 | 51.8 | 53.8 | 49.4 | 52.7 | 52.2 |
| Chen and Ng (2016) | 48.8 | 41.5 | 56.3 | **55.4** | 50.8 | 53.1 | 52.2 |
| Our Approach[†] | **59.2** | **51.3** | **60.5** | 53.9 | **55.5** | 52.9 | **55.3** |

Table 3: Experimental result (F-score) on the OntoNotes 5.0 test data. The best results are marked with bold face. † indicates that our approach is statistical significant over the baselines (using t-test, with $p < 0.05$). The number in the brackets indicate the number of AZPs.

see that our model significantly outperforms the previous state-of-the-art system (Chen and Ng, 2016) by 3.1% in overall F-score, and substantially outperform the other systems by a large margin. When observing the performances of different domains, our approach also gives relatively consistent improvements among various domains, except for BN and TC with a slight drop. All these results approve that our proposed approach is effective and achieves significant improvements in AZP resolution.

In our quantitative analysis, we investigated the reasons of the declines in the BN and TC domain. A primary observation is that the word distributions in these domains are fairly different from others. The average document length of BN and TC are quite longer than other domains, which suggest that there is a bigger chance to have unknown words than other domains, and add difficulties to the model training. Also, we have found that in the BN and TC domains, the texts are often in oral form, which means that there are many irregular expressions in the context. Such expressions add noise to the model, and it is difficult for the model to extract useful information in these contexts. These phenomena indicate that further improvements can be obtained by filtering stop words in contexts, or increasing the size of task-specific data, while we leave this in the future work.

- **Effect of UNK processing**

As we have mentioned in the previous section, traditional unknown word replacing methods are vulnerable to the real word test. To alleviate this issue, we proposed the UNK processing mechanism to recover the UNK tokens to the real words. In Table 4, we compared the performance that with and without the proposed UNK processing,

to show whether the proposed UNK processing method is effective. As we can see that, by applying our UNK processing mechanism, the model do learned the positional features in these low-frequency words, and brings over 3% improvements in F-score, which indicated the effectiveness of our UNK processing approach.

|  | F-score |
|---|---|
| Without UNK replacement | 52.2 |
| With UNK replacement | **55.3** |

Table 4: Performance comparison on whether using the proposed unknown words processing.

- **Effect of Domain Adaptation**

We also tested out whether our domain adaptation method is effective. In this experiments, we used three different types of training data: only pseudo training data, only task-specific data, and our adaptation method, i.e. using pseudo training data in the pre-training step and task-specific data for domain adaptation step. The results are given in Table 5. As we can see that, using either pseudo training data or task-specific data alone can not bring inspiring result. By adopting our domain adaptation method, the model could give significant improvements over the other models, which demonstrate the effectiveness of our proposed two-step training approach. An intuition behind this phenomenon is that though pseudo training data is fairly big enough to train a reliable model parameters, there is still a gap to the real zero pronoun resolution tasks. On the contrary, though task-specific training data is exactly the same type as the real test, the quantity is not enough to train a reasonable model (such as word embedding). So it is better to make use of both to

take the full advantage.

However, as the original task-specific data is fairly small compared to pseudo training data, we also wondered if the large-scale pseudo training data is only providing rich word embedding information. So we use the large-scale pseudo training data for embedding training using GloVe toolkit (Pennington et al., 2014), and initialize the word embeddings in the "only task-specific data" system. From the result we can see that the pseudo training data provide more information than word embeddings, because though we used GloVe embeddings in "only task-specific data", it still can not outperform the system that uses domain adaptation which supports our claim.

|                                   | F-score |
| --------------------------------- | ------- |
| Only Pseudo Training Data         | 41.1    |
| Only Task-Specific Data           | 44.2    |
| Only Task-Specific Data + GloVe   | 50.9    |
| Domain Adaptation                 | **55.3** |

Table 5: Performance comparison of using different training data.

## 4 Error Analysis

To better evaluate our proposed approach, we performed a qualitative analysis of errors, where two major errors are revealed by our analysis, as discussed below.

### 4.1 Effect of Unknown Words

Our approach does not do well when there are lots of ⟨unk⟩s in the context of ZPs, especially when the ⟨unk⟩s appears near the ZP. An example is given below, where words with # are regarded as ⟨unk⟩s in our model.

> $\phi$ 登上# 太平山# 顶，将 香港岛# 和 维多
> 利亚港# 的 美景 尽收眼底 。
> $\phi$ Successfully climbed# the peak of [Taiping Mountain]#, to have a panoramic view of the beauty of [Hong Kong Island]# and [Victoria Harbour]#.

In this case, the words "登上/climbed" and "太平山/Taiping Mountain" that appears immediately after the ZP "$\phi$" are all regarded as ⟨unk⟩s in our model. As we model the sequence of words by RNN, the ⟨unk⟩s make the model more difficult to capture the semantic information of the sentence, which in turn influence the overall performance. Especially for the words that are near

the ZP, which play important roles when modeling context information for the ZP. By looking at the word "顶/peak", it is hard to comprehend the context information, due to the several surrounding ⟨unk⟩s. Though our proposed unknown words processing method is effective in empirical evaluation, we think that more advanced method for unknown words processing would be of a great help in improving comprehension of the context.

### 4.2 Long Distance Antecedents

Also, our model makes incorrect decisions when the correct antecedents of ZPs are in long distance. As our model chooses answer from words in the context, if there are lots of words between the ZP and its antecedent, more noise information are introduced, and adds more difficulty in choosing the right answer. For example:

> 我 帮 不 了 那个 人 … … 那天 结束 后 $\phi$ 回到 家中 。
> I can't help that guy … … After that day, $\phi$ return home.

In this case, the correct antecedent of ZP "$\phi$" is the NP candidate "我/I". By seeing the contexts, we observe that there are over 30 words between the ZP and its antecedent. Although our model does not intend to fill the ZP gap only with the words near the ZP, as most of the antecedents appear just a few words before the ZPs, our model prefers the nearer words as correct antecedents. Hence, once there are lots of words between ZP and its nearest antecedent, our model can sometimes make wrong decisions. To correctly handle such cases, our model should learn how to filter the useless words and enhance the learning of long-term dependency.

## 5 Related Work

### 5.1 Zero pronoun resolution

For Chinese zero pronoun (ZP) resolution, early studies employed heuristic rules to Chinese ZP resolution. Converse (2006) proposes a rule-based method to resolve the zero pronouns, by utilizing Hobbs algorithm (Hobbs, 1978) in the CTB documents. Then, supervised approaches to this task have been vastly explored. Zhao and Ng (2007) first present a supervised machine learning approach to the identification and resolution of Chinese ZPs. Kong and Zhou (2010) develop a tree-kernel based approach for Chinese ZP resolution. More recently, unsupervised approaches

have been proposed. Chen and Ng (2014) develop an unsupervised language-independent approach, utilizing the integer linear programming to using ten overt pronouns. Chen and Ng (2015) propose an end-to-end unsupervised probabilistic model for Chinese ZP resolution, using a salience model to capture discourse information. Also, there have been many works on ZP resolution for other languages. These studies can be divided into rule-based and supervised machine learning approaches. Ferrández and Peral (2000) proposed a set of hand-crafted rules for Spanish ZP resolution. Recently, supervised approaches have been exploited for ZP resolution in Korean (Han, 2006) and Japanese (Isozaki and Hirao, 2003; Iida et al., 2006, 2007; Sasano and Kurohashi, 2011). Iida and Poesio (2011) developed a cross-lingual approach for Japanese and Italian ZPs where an ILP-based model was employed to zero anaphora detection and resolution.

In sum, most recent researches on ZP resolution are supervised approaches, which means that their performance highly relies on large-scale annotated data. Even for the unsupervised approach (Chen and Ng, 2014), they also utilize a supervised pronoun resolver to resolve ZPs. Therefore, the advantage of our proposed approach is obvious. We are able to generate large-scale pseudo training data for ZP resolution, and also we can benefit from the task-specific data for fine-tuning via the proposed two-step training approach.

### 5.2 Cloze-style Reading Comprehension

Our neural network model is mainly motivated by the recent researches on cloze-style reading comprehension tasks, which aims to predict one-word answer given the document and query. These models can be seen as a general model of mining the relations between the document and query, so it is promising to combine these models to the specific domain.

A representative work of cloze-style reading comprehension is done by Hermann et al. (2015). They proposed a methodology for obtaining large quantities of $\langle \mathcal{D}, \mathcal{Q}, \mathcal{A} \rangle$ triples. By using this method, a large number of training data can be obtained without much human intervention, and make it possible to train a reliable neural network. They used attention-based neural networks for this task. Evaluation on CNN/DailyMail datasets showed that their approach is much effective than traditional baseline systems.

While our work is similar to Hermann et al. (2015), there are several differences which can be illustrated as follows. Firstly, though we both utilize the large-scale corpus, they require that the document should accompany with a brief summary of it, while this is not always available in most of the document, and it may place an obstacle in generating limitless training data. In our work, we do not assume any prerequisite of the training data, and directly extract queries from the document, which makes it easy to generate large-scale training data. Secondly, their work mainly focuses on reading comprehension in the general domain. We are able to exploit large-scale training data for solving problems in the specific domain, and we proposed two-step training method which can be easily adapted to other domains as well.

## 6 Conclusion

In this study, we propose an effective way to generate and exploit large-scale pseudo training data for zero pronoun resolution task. The main idea behind our approach is to automatically generate large-scale pseudo training data and then utilize an attention-based neural network model to resolve zero pronouns. For training purpose, two-step training approach is employed, i.e. a **pre-training** and **adaptation** step, and this can be also easily applied to other tasks as well. The experimental results on OntoNotes 5.0 corpus are encouraging, showing that the proposed model and accompanying approaches significantly outperforms the state-of-the-art systems.

The future work will be carried out on two main aspects: First, as experimental results show that the unknown words processing is a critical part in comprehending context, we will explore more effective way to handle the UNK issue. Second, we will develop other neural network architecture to make it more appropriate for zero pronoun resolution task.

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
