# Peer review of "Generating and Exploiting Large-scale Pseudo Training Data for Zero Pronoun Resolution"

_ACL 2017 — decision unknown_

[Official Review · Reviewer 1 · rating 4 · confidence 4]
soundness 5 · originality 3 · clarity 4 · substance 3 · appropriateness 5 · presentation format Oral Presentation

- Strengths:
This paper introduced a novel method to improve zero pronoun resolution
performance.. The main contributions of this papers are: 1) proposed a simple
method to automatically generate a large training set for zero pronoun
resolution task; 2) adapted a two step learning process to transfer knowledge
from large data set to the specific domain data; 3) differentiate unknown words
using different tags. In general, the paper is well written. Experiments are
thoroughly designed. 

- Weaknesses:

But I have a few questions regarding finding the antecedent of a zero pronoun:
1. How will an antecedent be identified, when the prediction is a pronoun? The
authors proposed a method by matching the head of noun phrases. It’s not
clear how to handle the situation when the head word is not a pronoun.
2. What if the prediction is a noun that could not be found in the previous
contents?
3. The system achieves great results on standard data set. I’m curious is it
possible to evaluate the system in two steps? The first step is to evaluate the
performance of the model prediction, i.e. to recover the dropped zero pronoun
into a word; the second step is to evaluate how well the systems works on
finding an antecedent.

I’m also curious why the authors decided to use attention-based neural
network. A few sentences to provide the reasons would be helpful for other
researchers.

A minor comment:
In figure 2, should it be s1, s2 … instead of d1, d2 ….? 

- General Discussion:
Overall it is a great paper with innovative ideas and solid experiment setup.

[Official Review · Reviewer 2 · rating 4 · confidence 3]
soundness 5 · originality 3 · clarity 4 · substance 4 · appropriateness 5 · presentation format Poster

- Strengths:

The approach is novel and the results are very promising, beating
state-of-the-art.

- Weaknesses:

 The linguistic motivation behind the paper is troublesome (see below). I feel
that the paper would benefit a lot from a more thoughtful interpretation of the
results.

- General Discussion:

This paper presents an approach for Zero Pronoun Resolution in Chinese. The
authors advocate a novel procedure for generating large amount of relevant data
from unlabeled documents. These data are then integrated smartly in an NN-based
architecture at a pre-training step. The results improve on state-of-the-art.

I have mixed feelings about this study. On the one hand, the approach seems
sound and shows promising results, beating very recent systems (e.g., Chen&Ng
2016). On the other hand, the way the main contribution is framed is very
disturbing from the linguistic point of view. In particular, (zero) pronoun
resolution is, linguistically speaking, a context modeling task, requiring
accurate interpretation of discourse/salience, semantic and syntactic clues. It
starts from the assumption that (zero) pronouns are used in specific contexts,
where full NPs shouldn't normally be possible. From this perspective,
generating ZP data via replacing nominal with zeroes ("blank") doesn't sound
very convincing. And indeed, as the authors themselves show, the pre-training
module alone doesn't achieve a reasonable performance. To sum it up, i don't
think that these generated pseudo-data can be called AZP data. It seems more
likely that they encode some form of selectional preferences (?). It would be
nice if the authors could invest some effort in better understanding what
exactly the pre-training module learns -- and then reformulate the
corresponding sections. 

The paper can benefit from a proofreading by a native speaker of English -- for
example, the sentence on lines 064-068 is not grammatical.

-- other points --

lines 78-79: are there any restrictions on the nouns and especially pronouns?
for example, do you use this strategy for very common pronouns (as English
"it")? if so, how do you guarantee that the two occurrences of the same token 
are indeed coreferent?

line 91: the term antecedent is typically used to denote a preceding mention
coreferent with the anaphor, which is not what you mean here

line 144: OntoNotes (typo)

lines 487-489: it has been shown that evaluation on gold-annotated data does
not provide reliable estimation of performance. and, indeed, all the recent
studies of coreference evaluate on system mentions. for example, the studies of
Chen&Ng you are citing, provide different types of evaluation, including those
on system mentions. please consider rerunning your experiments to get a more
realistic evaluation setup

line 506: i don't understand what the dagger over the system's name means. is
your improvement statistically significant on all the domains? including bn and
tc??

line 565: learn (typo)

section 3.3: in this section you use the abbreviation AZP instead of ZP without
introducing it, please unify the terminology

references -- please double-check for capitalization